# Spatial-Aware Transformer (SAT): Enhancing Global Modeling in Transformer Segmentation for Remote Sensing Images

Duolin Wang [1,2], Yadang Chen [1,2], Bushra Naz [3], Le Sun [1,2,4] and Baozhu Li [5,*]

1    School of Computer Science, Nanjing University of Information Science and Technology,
     Nanjing 210044, China; 20211221049@nuist.edu.cn (D.W.); adamchen@nuist.edu.cn (Y.C.);
     sunlecncom@nuist.edu.cn (L.S.)
2    Engineering Research Center of Digital Forensics, Ministry of Education, Nanjing University of Information
     Science and Technology, Nanjing 210044, China
3    Department of Computer Systems Engineering, Mehran University of Engineering and Technology,
     Jamshoro 76062, Pakistan; bushra.naz@faculty.muet.edu.pk
4    Jiangsu Collaborative Innovation Center of Atmospheric Environment and Equipment Technology
     (CICAEET), Nanjing University of Information Science and Technology, Nanjing 210044, China
5    Internet of Things & Smart City Innovation Platform, Zhuhai Fudan Innovation Institute,
     Zhuhai 519031, China
*    Correspondence: baozhuli@fudan-zhuhai.org.cn

**Abstract:** In this research, we present the Spatial-Aware Transformer (SAT), an enhanced implementation of the Swin Transformer module, purposed to augment the global modeling capabilities of existing transformer segmentation mechanisms within remote sensing. The current landscape of transformer segmentation techniques is encumbered by an inability to effectively model global dependencies, a deficiency that is especially pronounced in the context of occluded objects. Our innovative solution embeds spatial information into the Swin Transformer block, facilitating the creation of pixel-level correlations, and thereby significantly elevating the feature representation potency for occluded subjects. We have incorporated a boundary-aware module into our decoder to mitigate the commonly encountered shortcoming of inaccurate boundary segmentation. This component serves as an innovative refinement instrument, fortifying the precision of boundary demarcation. After these strategic enhancements, the Spatial-Aware Transformer achieved state-of-the-art performance benchmarks on the Potsdam, Vaihingen, and Aerial datasets, demonstrating its superior capabilities in recognizing occluded objects and distinguishing unique features, even under challenging conditions. This investigation constitutes a significant advancement toward optimizing transformer segmentation algorithms in remote sensing, opening a wealth of opportunities for future research and development.

**Keywords:** image segmentation; self-attention; semi-supervised learning; self-attention mechanism for remote sensing

## 1. Introduction

The utility and prominence of image segmentation algorithms in the remote sensing [1,2] domain are well established. The processing and interpretation of satellite and aerial imagery have been drastically transformed, unlocking a diverse array of applications. This has allowed for more nuanced environmental monitoring, detailed urban planning, and intricate military surveillance, among other applications. As the need for high-resolution feature extraction and object identification becomes increasingly pressing, so does the importance of refining and improving these segmentation algorithms.

Fully Convolutional Networks (FCNs) [3], U-Net [4], and SegNet [5] stand as noteworthy examples among the plethora of image segmentation algorithms employed. FCNs [3,6,7] are prized for their ability to handle varying input sizes, an attribute that has led to admirable segmentation results across several scenarios. However, FCNs' scope in extracting

global features is limited. This drawback often becomes evident in tasks that require a broader contextual understanding. On the other hand, U-Net-based methods [4,8,9] boast an efficient encoder–decoder architecture, which has been particularly successful in biomedical image segmentation. Yet, their application in remote sensing is constrained due to their limited capacity to manage large-scale, high-resolution images. SegNet [5,10] demonstrates proficiency in segmenting complex scenes but exhibits shortcomings in accurately delineating boundaries and effectively handling occluded objects' essential capabilities in remote sensing image segmentation. The advent of Transformer-based models has revolutionized numerous fields, primarily due to their ability to model long-range dependencies without the constraints of local receptive fields, as is the case with CNNs [11,12]. Swin Transformers [13], a variant of the Transformer family [14–17], offer an additional advantage by enabling both local representations. However, they limit global modeling capabilities in the Transformer-based methods.

To address these limitations, we have developed the Spatial-Aware Transformer, a constituent of the Swin Transformer block, to optimize the employment of self-attention strategies, forge pixel-level correspondences, and augment the faculty of feature representation. As Figure 1a shows, our segmentation algorithm may encounter challenges in accurately identifying certain areas within the image. For instance, in the white box, the algorithm faces difficulty in distinguishing between an impervious surface and a car within the local region. Similarly, as illustrated by the yellow box, the algorithm struggles to differentiate between regions that belong to a car and those that are part of a building. As Figure 1b shows, the integration of spatial data within the Swin Transformer block via the Spatial-Aware component engenders a nuanced comprehension of pixel interrelationships, thereby facilitating improved segmentation accuracy. This approach equips the model with the capability to take into account not only the immediate local context but also the wider spatial context, thereby aiding in modeling obscured entities and extracting complex features. The integration of the Spatial-Aware component into our methodological approach yields several benefits. It enhances the feature representation capacity, equipping the model with the ability to capture the subtleties and complex structures present in remote sensing imagery with greater finesse. Furthermore, the module augments the model's capacity to model global dependencies through the utilization of the self-attention mechanism. This allows the model to perceive the spatial interrelations between objects, resulting in precise segmentation.

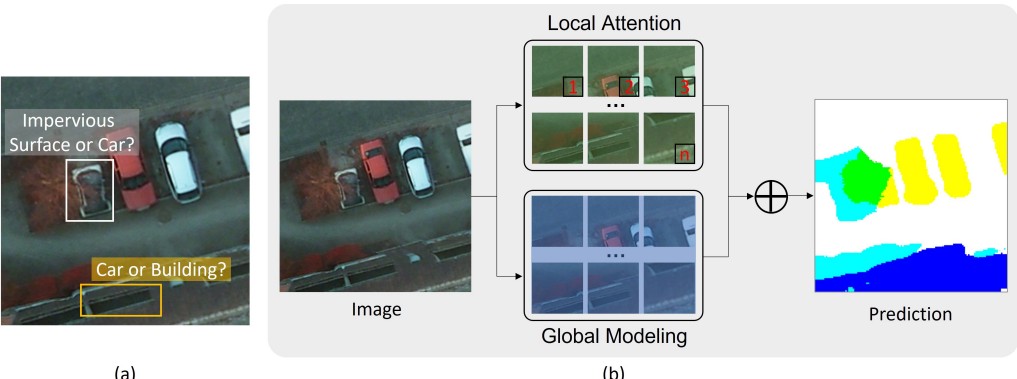

(a)    (b)

**Figure 1.** (**a**) Some problems that our segmentation algorithm may encounter are, as shown in the white box, it is difficult for the algorithm to determine whether the area is an impervious surface or a car in the local range, and as shown by the yellow box, it is difficult for the algorithm to distinguish whether the area is part of a car or a building. (**b**) The summary diagram of our method, the Swin Transformer segmentation picture ignores the importance of global modeling, there is almost no information exchange between blocks, and we propose a Spatial-Aware Transformer, which focuses on both the attention segmentation of a single small block and global modeling.

Another critical challenge that plagues remote sensing image segmentation pertains to the accuracy of edge or boundary segmentation. The complexity of remote sensing imagery is often exacerbated due to the top-view perspective of many images in datasets. Such a perspective can obscure the clear demarcation between objects, leading to inaccurate segmentation outputs, particularly around the objects' boundaries. It is essential to address this limitation, given the significance of precise boundary segmentation to the overall accuracy and utility of the segmentation task. We have integrated a boundary-aware module into our decoder, serving as a refinement tool that markedly improves boundary delineation. To this end, we propose the integration of a boundary-aware module into our decoder. This module brings about substantial improvements in boundary segmentation and functions as a refinement tool. The boundary-aware module exhibits a heightened sensitivity to boundary structures, which leads to more precise delineation of segmented regions. Consequently, the characterization of objects within the scene is significantly enhanced, contributing to more accurate segmentation outputs. Furthermore, our model employs a hybrid loss function, incorporating Binary Cross-Entropy (BCE) loss, Structural Similarity Index (SSIM) loss, and Intersection over Union (IoU) loss. This enables the model to learn from different aspects of the image.

Experimental evaluations of our proposed modules have exhibited impressive results and achieved state-of-the-art performance on three renowned datasets: Potsdam, Vaihingen, and Aerial. These results testify to the SAT's capability to effectively manage occlusion, delineate intricate features, and precisely segregate object boundaries, outperforming existing models.

To sum up, this study presents a pioneering approach toward refining image segmentation in remote sensing. By strategically addressing recognized limitations in current methods and incorporating unique, targeted enhancements, we offer an innovative solution that not only meets the field's current needs but also lays a strong foundation for future advancements.

## 2. Related Work

### 2.1. CNN-Based Remote Sensing Image Segmentation

Convolutional Neural Networks (CNNs) [11,12,18,19] have long been recognized as powerful tools for image analysis due to their ability to learn hierarchical feature representations from raw image data. They have been particularly successful in various segmentation tasks, thanks to their robustness in extracting spatial features from images.

Early works applied traditional CNN architectures, such as LeNet [20] and AlexNet [21], to remote sensing image segmentation tasks. These initial efforts demonstrated the potential of CNNs in this domain, revealing substantial improvements over traditional machine learning methods. However, the relatively shallow architectures of these initial CNNs were limited in their ability to capture complex patterns and structures inherent in remote sensing imagery.

The advent of deeper architectures, such as VGGNet [22] and ResNet [23], brought about significant improvements. These deep CNNs, armed with an increased number of layers, improved the capacity to capture more complex features and patterns within the remote sensing images. Particularly, ResNet introduced the concept of skip connections, which mitigated the vanishing gradient problem, thereby allowing for the effective training of deep networks.

Although deep CNNs offered substantial improvements over their predecessors, they were primarily constrained by their limited receptive fields, inhibiting their global modeling capabilities. In the context of remote sensing images, which often encompass intricate spatial structures and long-range dependencies, this limitation becomes particularly significant.

A range of methods have been proposed to tackle this issue. One notable approach is the integration of multi-scale features, as demonstrated in the Pyramid Scene Parsing Network (PSPNet) [24] and the Deeplab family of models [25–27]. These architectures em-

ploy spatial pyramid pooling and dilated convolutions, respectively, to capture contextual information at various scales.

Despite these advancements, conventional CNN-based segmentation algorithms still struggle to fully capture the complex spatial relationships and global context inherent in remote sensing imagery. This limitation underscores the need for new approaches, particularly those leveraging Transformer-based models, as explored in the next section.

### 2.2. Remote Sensing Image Segmentation Based on Self-Attention Mechanisms

The self-attention mechanism, also known as the Transformer, was introduced by [28]. In the domain of natural language processing, unlike CNNs, Transformers are not restricted by local receptive fields and are capable of capturing long-range dependencies in data, making them a promising solution for the global modeling challenge in remote sensing image segmentation.

The self-attention mechanism computes the response at a position as a weighted sum of the features at all positions in the data. This global context-awareness allows the Transformer to better capture intricate spatial structures and long-range dependencies that are characteristic of remote sensing images [19,29–31].

Initial applications of the Transformer in remote sensing image segmentation used hybrid models, combining CNNs for local feature extraction and Transformers for global context aggregation [32]. Examples include the Vision Transformer (ViT) [33] and the TransUNet [34]. These models achieved promising results, underscoring the potential of Transformers in this domain.

Despite their merits, the application of traditional Transformers to remote sensing image segmentation is not straightforward due to their high computational cost and the requirement for large-scale datasets for effective training [35]. To address this, researchers introduced the Swin Transformer [13], a variant that allows for both local and global representations, making it more suitable for segmentation tasks.

The Swin Transformer divides the input image into non-overlapping patches and processes them in a hierarchical manner, thus effectively capturing spatial information at multiple scales [13]. This characteristic is particularly advantageous in handling the complex spatial structures of remote sensing images.

Our research builds upon the aforementioned works, seeking to further advance the field of remote sensing image segmentation. We extend the Swin Transformer model by incorporating spatial awareness, enhancing its capacity to handle intricate and interconnected objects in remote sensing imagery. Additionally, we introduce a boundary-aware module to improve edge segmentation, addressing a critical challenge in this domain.

In summary, the research landscape of remote sensing image segmentation reveals significant progress, with CNN-based models providing the initial groundwork and Transformer-based models introducing exciting new possibilities. Our work seeks to contribute to this ongoing progress, offering a novel and comprehensive approach to enhance global modeling abilities and improve boundary segmentation accuracy.

### 3. Methods

This section outlines the details of the methods adopted in this research, specifically focusing on two primary components: the Spatial-Aware Transformer module, which enhances the global modeling capabilities of our segmentation algorithm, and the Boundary-Aware Refinement module, which sharpens the boundary segmentation in our output. These modules operate in conjunction and facilitate a novel and effective approach to remote sensing image segmentation.

### 3.1. Network Architecture

Figure 2 presents the comprehensive structure of our algorithm, melding U-Net's straightforward yet sophisticated traits. This is achieved by employing a skip connection

layer to bridge the encoder and decoder and integrating two primary components: the SAT and the mask refinement module.

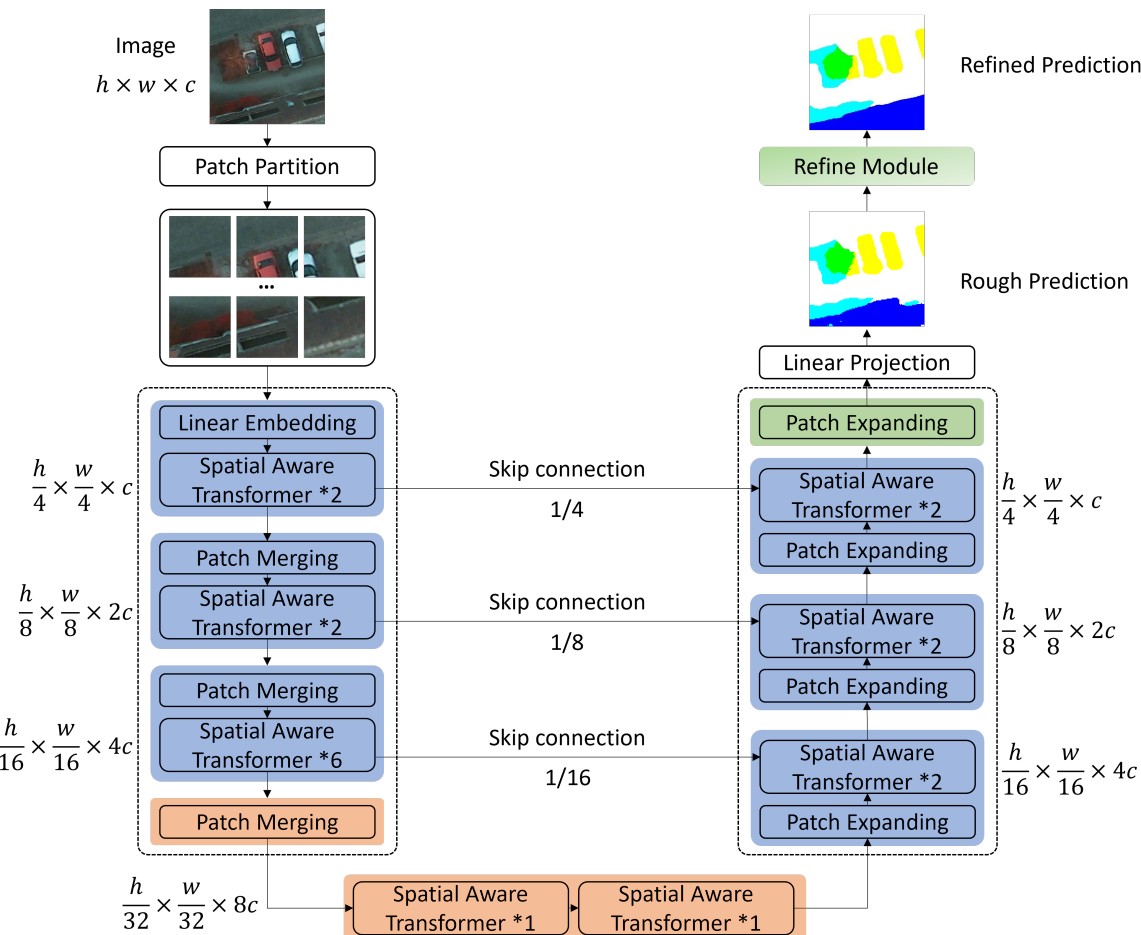

**Figure 2.** Architecture of our proposed method, which contains two important modules: Spatial-Aware Transformer, and Boundary-Aware Refinement module. The input image is divided into *n* small patches and processed through the encoder, which includes several spatial-aware transformer modules. The encoder's output is then fed into the bottleneck layer to extract deeper-level feature information, and the output of the bottleneck layer is passed to the decoder. The rough predicted mask from the decoder is input into the refine module to obtain the edge-optimized predicted mask.

In the encoder segment, the initial step involves taking the remote sensing image slated for division as the input and dissecting it into non-overlapping segments, each of which we consider as a token. These tokens are not inherently connected in the context of the language model. However, in the Vision Transformer (ViT), not only are the segments interconnected, but the pixels within these segments also share robust linkages.

To augment the semantic correlation among pixels, we establish the segment size at $8 \times 8$ and set the overlapping percentage to 50%. Subsequently, each divided segment undergoes a Linear Embedding process, acquiring the feature of the c-dimension. This feature is then inserted into the Spatial-Aware Transformer module, effectively mitigating the constraint of window self-attention, allowing for global pixel-level information pairing, and reducing the semantic uncertainty induced by obstruction.

Next, the encoder's output is routed into the bottleneck layer, constructed from a pair of Spatial-Aware Transformer modules, without altering the feature dimension. This process enables the maximum possible exploration within the features. The resultant output from the bottleneck layer is then fed into the decoder layer to generate a preliminary

prediction mask. Then, this mask is processed through the Boundary-Aware module to produce the final refined prediction mask.

### 3.2. Transform Module with Spatial Awareness

The basis of our Spatial-Aware Transformer module is the Swin Transformer, a specialized variant of the generic Transformer model, which has seen significant success in a myriad of visual tasks. We first delineate the fundamental principles behind the Swin Transformer before delving into the specifics of our proposed module.

#### 3.2.1. Swin Transformer

The distinctive feature of the Swin Transformer is its "shifted window" approach to partitioning an input image. The process commences by dividing the input image into miniature patches, each measuring 4 px by 4 px. Each of these patches has three channels, summing up to 48 feature dimensions ($4 \times 4 \times 3 = 48$).

These patches, each of 48 dimensions, are subsequently linearly transformed into a dimensionality denoted as $C$, effectively converting these patches into vectors of dimension $C$. The choice of $C$ is a design choice that impacts the size of the Transformer model and, consequently, the number of hidden parameters in the model's fully connected layers.

The Swin Transformer revolutionizes image processing by transitioning from the Vision Transformer's quadratic computational approach to a more streamlined linear complexity. This is achieved by focusing self-attention within local windows rather than globally, making the Swin Transformer more adept at dense recognition tasks and versatile in remote sensing applications.

Moving on, as depicted in Figure 3, the blue box signifies the Swin Transformer's primary operation. We designate the input as 'I'. Prior to feeding 'I' into the transformer, an initial patch merging is executed, which serves the dual purpose of downsampling and setting the stage for hierarchical structure formation. This not only minimizes resolution but optimizes the channel count, conserving computational resources. In particular, each downsampling doubles, selecting elements at intervals of two both in rows and columns of the input feature 'I', which are then amalgamated into a single tensor. At this stage, the channel dimension increases 4-fold, which is subsequently fine-tuned to twice its original size via a fully connected layer.

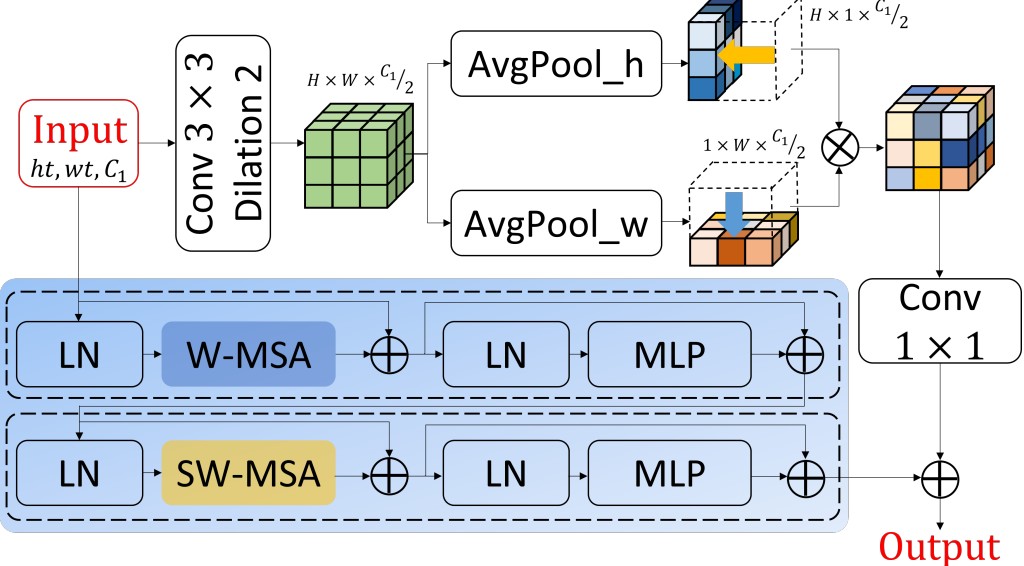

**Figure 3.** Architecture of the Spatial-Aware Transformer module.

Diving into specifics, the output from the fully connected layer is input into the transformer module. Here, post layer normalization, Window Self-Attention (Win-SA)

is performed. The resultant attention-modulated features are then passed through another layer normalization layer followed by a Multilayer Perception (MLP) layer. The mathematical representation is as follows:

$$\hat{\mathbf{f}}^t = \text{Win} - \text{SA}\left(\text{LNorm}\left(\mathbf{f}^{t-1}\right)\right) + \mathbf{f}^{t-1}$$
$$\mathbf{f}^t = \text{MLP}\left(\text{LNorm}\left(\hat{\mathbf{f}}^t\right)\right) + \hat{\mathbf{f}}^t \tag{1}$$

where $\mathbf{f}^{t-1}$ denotes the input feature of the transformer module, 'LNorm' signifies layer normalization, and 'Win-SA' represents Window Multi-head Self-Attention. The term $\hat{\mathbf{f}}^t$ reflects the sum of the output feature from Window Self-Attention and the input feature. At this juncture, input image features are segmented into $W \times W$ blocks, with each block treated as a window where self-attention is performed. This strategy effectively mitigates the computational strain of the Vision Transformer, which can struggle with handling large image features. Additionally, the window size is a flexible hyper-parameter that is typically chosen based on computational efficacy and task demands. In this experiment, we chose $W$ as 8 for ease of feature computation and practicality.

However, partitioning image features into smaller windows comes at the expense of the self-attention mechanism's core premise, wherein any feature point should be able to communicate with others. The fact that attention is restricted within windows results in a lack of inter-window information exchange. To counter this, a window shift operation is employed, involving passing the $\mathbf{f}^{t-1}$ feature through the layer normalization layer and feeding it into the Shifted Window Self-Attention (SWin-SA) module. Thereafter, the attention-matched feature is passed through another layer normalization layer and an MLP layer. This process can be mathematically represented as follows:

$$\hat{\mathbf{f}}^{t+1} = \text{SWin} - \text{SA}\left(\text{LNorm}\left(\mathbf{f}^t\right)\right) + \mathbf{f}^t$$
$$\mathbf{f}^{t+1} = \text{MLP}\left(\text{LNorm}\left(\hat{\mathbf{f}}^{t+1}\right)\right) + \hat{\mathbf{f}}^{t+1} \tag{2}$$

where $\hat{\mathbf{f}}^{t+1}$ indicates the combined output features and input feature after the Shifted Window Self-Attention operation, and $\mathbf{f}^{t+1}$ represents the combined output of the MLP layer and $\hat{\mathbf{f}}^{t+1}$. This step helps ensure that the essential aspect of self-attention—interaction between different feature points—is retained.

### 3.2.2. Spatial-Aware Transformer

As we pivot to the Spatial-Aware Transformer, it is important to recognize a trade-off with the Swin Transformer block. Although it efficiently establishes relationships among patch tokens within a bounded window and reduces memory consumption, there is a downside: it marginally impedes the Transformer's ability to model global relationships. This constraint remains despite the use of alternating standard and shifted windows.

Delving into the specifics, in remote sensing imagery, the obscuring of ground objects frequently leads to indistinct boundaries. Addressing this requires the integration of spatial information for sharpening the edges. Hence, as a solution, we introduce the Spatial-Aware Transformer (SAT) in alliance with the W-Transform and SW-Transform blocks. This integration facilitates enriched information exchange and encodes more nuanced spatial details.

Notably, SAT distinguishes itself by employing attention mechanisms across two spatial dimensions. In doing so, it focuses on the interrelationships between individual pixels, rather than solely on patch tokens. This adaptation makes the transformer particularly adept at handling image segmentation tasks. For visual clarity, the constituents of SAT are graphically represented in Figure 3.

During stage $m$, the W-Transform block takes an input feature $\mathbf{f}^{t-1} \in \mathbb{R}^{(h \times w) \times c_1}$ and reshapes it into $\mathbf{s} \in \mathbb{R}^{h \times w \times c_1}$. For clarity, let's define $c_1 = 2^{m-1}C_1$, $ht = \left(H/\left(2^{m+1}\right)\right)$, and $wt = \left(W/\left(2^{m+1}\right)\right)$. The next step involves processing feature $\mathbf{s}$ through a $3 \times 3$

dilated convolution layer with a dilation rate of 2. This operation revitalizes the structural information of the feature map by expanding its receptive field and, at the same time, reduces the number of channels to $c_1/2$ for the sake of efficiency.

Following this, global average pooling comes into play to distill the spatial statistics in both vertical and horizontal directions from the feature map. To provide more detail on this aspect, the calculation for elements in each direction is articulated as follows:

$$
\begin{aligned}
v_{ht_i}^k &= \frac{1}{w} \sum_{j=0}^{wt-1} \hat{\mathbf{s}}_2^c(i,j) \\
v_{wt_j}^k &= \frac{1}{h} \sum_{i=0}^{ht-1} \hat{\mathbf{s}}_2^c(i,j)
\end{aligned}
\tag{3}
$$

where $i$, $j$, and $c_2$ act as indices for the vertical and horizontal directions, and the channel, respectively, where $0 \leq i < ht, 0 \leq j < wt, 0 \leq k < c_1/2$. We formulate the feature $\hat{\mathbf{s}}$ as $f(\mathbf{s})$, with $f(\cdot)$ denoting a dilated convolution layer, incorporating batch normalization and the GELU activation function. The accumulated tensors in the vertical and horizontal directions, calculated as per Equation (3), are represented as $\mathbf{v}_h t \in \mathbb{R}^{ht \times 1 \times (c_1/2)}$ and $\mathbf{v}_w t \in \mathbb{R}^{1 \times w \times (c_1/2)}$, respectively.

The tensors $\mathbf{v}_h t$ and $\mathbf{v}_w t$ merge the pixel-level weights of the feature map spatially. The resulting product forms the position-aware attention map $\mathbf{A_M}$, defined as $\mathbf{A_M} \in \mathbb{R}^{ht \times wt \times (c_1/2)}$. Subsequently, the output feature map $\mathbf{F_R}$ of SAT is achieved by combining $\mathbf{A_M}$ and the output of the SWin-Transform block, denoted as $\mathbf{f}^{t+1}$. It is vital to remember that a convolutional layer enlarges the dimensions of $\mathbf{A_M}$ to align with the dimensions of feature $\mathbf{f}^{t+1}$. Thus, feature $\mathbf{F_R} \in \mathbb{R}^{ht \times wt \times c_1}$ emerges, as depicted below:

$$
\mathbf{F_R} = \mathbf{f}^{t+1} \oplus \varphi(\mathbf{v}_h t \otimes \mathbf{v}_w t)
\tag{4}
$$

where "$\otimes$" represents matrix multiplication and "$\oplus$" is used to denote element-wise addition. Furthermore, "$\varphi(\cdot)$" is indicative of a $1 \times 1$ convolutional layer, which employs batch normalization coupled with the GELU activation function.

In summary, the Swin Transformer block, with its local window-based self-attention operation and accompanying MLP, provides an efficient and effective method for handling image data within the Transformer model context. Its design and operations have made significant contributions to the field of remote sensing image segmentation tasks.

### 3.3. Boundary-Aware Refinement Module

To address the challenge of accurate boundary segmentation in Spatial-Aware Transformers, we introduce a two-step modification: integrating a bottleneck layer behind the encoder and inputting the coarse prediction mask into a boundary refinement module.

#### 3.3.1. Bottleneck Layer Integration

The Bottleneck Layer Integration module is designed to leverage the hierarchical representation power of the Swin Transformer and enhance its segmentation performance. By incorporating two consecutive Swin Transformer modules as the bottleneck layer, we enable the model to capture more complex spatial dependencies and obtain deeper contextual information.

Let $X \in \mathbb{R}^{H \times W \times C}$ denote the input feature map to the Bottleneck Layer Integration module, where $H$ represents the height, $W$ represents the width, and $C$ represents the number of channels. The feature map $X$ is processed by the first Swin Transformer module, denoted as $f_{\text{Swin}}^{(1)}$. This module applies self-attention mechanisms and feed-forward networks to refine the feature representation. The output of $f_{\text{Swin}}^{(1)}$ is denoted as $Y \in \mathbb{R}^{H \times W \times C'}$, where $C'$ represents the number of output channels.

The output feature map $Y$ is then fed into the second Swin Transformer module, denoted as $f_{\text{Swin}}^{(2)}$. This module further refines the feature representation by applying self-attention mechanisms and feed-forward networks. The output of $f_{\text{Swin}}^{(2)}$ is denoted as $Z \in \mathbb{R}^{H \times W \times C''}$, where $C''$ represents the final number of output channels.

To integrate the bottleneck layer into the overall segmentation architecture, the output feature map $Z$ is concatenated with the input feature map $X$. The concatenation operation is performed along the channel dimension, resulting in a fused feature map $X \oplus Z \in \mathbb{R}^{H \times W \times (C+C'')}$.

To ensure the consistency of feature dimensions throughout the network, a $1 \times 1$ convolutional layer is applied to $X \oplus Z$ to adjust the number of channels back to the original value. The output of this convolutional layer is denoted as $B \in \mathbb{R}^{H \times W \times C}$, representing the integrated bottleneck layer.

The Bottleneck Layer Integration module enriches the segmentation model with deeper contextual information and enhances its ability to capture complex spatial dependencies. By employing two consecutive Swin Transformer modules as the bottleneck layer, our model achieves improved segmentation performance and higher-level feature representation.

### 3.3.2. Refinement Module

As Figure 4 shows, the refinement module is comprised of several components: a $3 \times 3$ convolutional layer, batch normalization, a ReLU activation function, and a max-pooling operation. Each of these components is integrated into every convolutional kernel layer during the encoding phase. Transitioning to the decoding phase, the model employs bilinear interpolation as its upsampling technique. Post-upsampling, convolutional layers and long skip connections are strategically implemented. These connections bridge the respective decoders and encoders, thereby facilitating the incorporation of supplemental contextual information. This streamlined structure ensures efficiency while preserving the critical attributes of the data through various stages of processing.

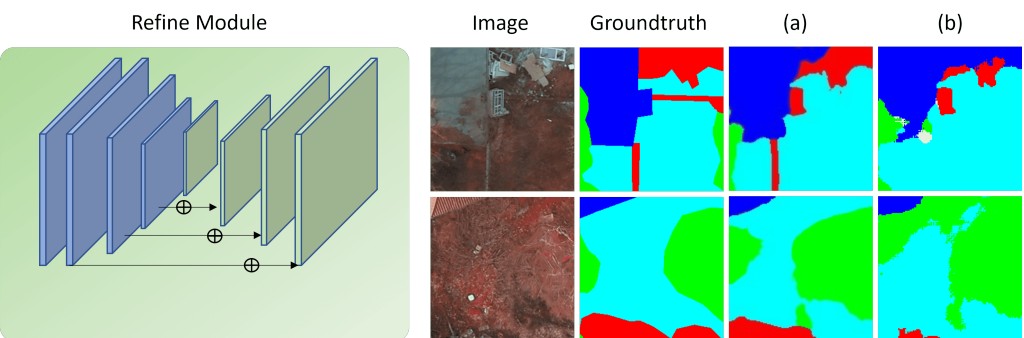

**Figure 4.** On the left is a schematic diagram of the refinement module. On the right is the segmented contrast image with and without the refinement module, (**a**) represents the segmented image with the refinement module, and (**b**) represents the comparison image without the refinement module. Different colors in the image represent different segmented objects.

### 3.3.3. Conjoint Loss Function

The model utilizes a conjoint loss function, incorporating BCE loss, SSIM loss, and IoU loss. This allows the model to glean insights from various facets of the image. The hybrid loss function is defined as follows:

$$LOSS = \alpha LOSS_{BCE}(U, \hat{U}) + \beta LOSS_{SSIM}(U, \hat{U}) + \gamma LOSS_{IOU} \tag{5}$$

In the given formulation, we have $LOSS_{BCE}$, $LOSS_{SSIM}$, and $LOSS_{IOU}$ representing the BCE loss, SSIM loss, and IoU loss, respectively. The ground truth is denoted by $U$, the predicted output is $\hat{U}$, and $\alpha$, $\beta$, and $\gamma$ are the weights assigned to each loss component.

**Binary Cross-Entropy Loss**

The BCE loss function is extensively employed for binary classification tasks. It assesses the performance of a classification model that generates probability values ranging from 0 to 1. The BCE loss is defined as:

$$LOSS_{BCE}(U, \hat{U}) = 1 - (U \cdot \log(\hat{U}) + (1 - U) \cdot \log(1 - \hat{U})) \tag{6}$$

where $U$ represents the actual label, $\hat{U}$ represents the predicted label.

**Structural Similarity Index Loss**

The Structural Similarity Index (SSIM) loss function is a perceptual loss function employed to quantify the similarity between two images. It compares the local patterns of normalized pixel intensities and is commonly employed as a metric for image similarity. The SSIM index is computed using image windows:

$$LOSS_{SSIM}(U, \hat{U}) = 1 - \frac{(2\mu_U \mu_{\hat{U}} + X)(2\sigma_{U\hat{U}} + Y)}{(\mu_U^2 + \mu_{\hat{U}}^2 + X)(\sigma_U^2 + \sigma_{\hat{U}}^2 + Y)} \tag{7}$$

Here, $U$ and $\hat{U}$ represent the ground truth image and the predicted (reconstructed) image, respectively, $\mu_U$ and $\mu_{\hat{U}}$ denote their respective averages, $\sigma_{U\hat{U}}$ represents the covariance between $U$ and $\hat{U}$, and $X$ and $Y$ are variables employed to stabilize the division with a weak denominator.

**Intersection over Union (IoU) Loss**

The IoU loss measures the overlap between two bounding boxes and can be employed as a loss function for object detection tasks. The IoU is computed by dividing the area of intersection between two bounding boxes by the area of their union, as follows:

$$LOSS_{IOU} = 1 - \frac{\text{Ar\_intersect}}{\text{Ar\_union}} \tag{8}$$

where we denote the intersection area as "Ar_intersect" and the union area as "Ar_union".

In conclusion, the proposed method produces an optimized prediction mask, demonstrating improved boundary segmentation performance. This highlights the novel application of the Spatial-Aware Transformer in addressing existing challenges in remote sensing image segmentation.

## 4. Experiments

### *4.1. Datasets*

Our study was validated on the basis of two types of datasets. First, the COCO dataset, known for its variety and comprehensive feature set. The COCO dataset offers us a multitude of annotated everyday objects, facilitating the early-stage learning of distinctive features. Subsequently, we employ three remote sensing datasets, Vaihingen, Potsdam, and Aerial, each providing high-resolution imagery capturing diverse landscapes.

#### 4.1.1. Vaihingen Dataset

The Vaihingen dataset is provided for the semantic segmentation of urban scenes. It contains aerial images of Vaihingen along with ground truth labels. The on-board scanner provides these images with very high resolution, which makes them a good choice for evaluating the performance of segmentation algorithms. The Vaihingen dataset includes several different types of urban features. The high resolution of the images (approximately 9 cm per pixel) enables the detection and classification of these classes at a high level of detail. The Vaihingen dataset has been used in numerous studies on semantic segmentation and other related tasks in remote sensing.

### 4.1.2. Aerial Imagery Datasets

This dataset comprises images captured from an airborne perspective, such as from a drone, plane, or satellite. Aerial image datasets are widely used in the computer vision field to facilitate a variety of tasks like image segmentation, object detection, and scene recognition. They are key to many applications, including land cover mapping, urban planning, environmental monitoring, disaster management, and military reconnaissance. Aerial images present unique challenges due to their high-resolution, large coverage area, and variations in view angles, lighting conditions, and land cover types.

### 4.1.3. Potsdam Dataset

The Potsdam dataset is a popular benchmark dataset provided for 2D Semantic Labeling. The ground sampling distance is approximately 5 cm, offering high detail. The dataset provides corresponding ground truth images where each pixel has been labelled with one of six categories, as shown in Figure 5. In the course of our experimental process, each image within the dataset is constituted by three distinct channel combinations: namely, Infrared-Red-Green (IR-R-G), Red-Green-Blue (R-G-B), and a four-channel variant Red-Green-Blue-Infrared (R-G-B-IR). We employed a dataset comprised of 17 images for training, each possessing the R-G-B channel combination. The remaining seven R-G-B images were assigned for the testing. For both training and testing, the original images were uniformly cropped to patches of 256 × 256 pixels to maintain consistency in the data handling and processing workflow.

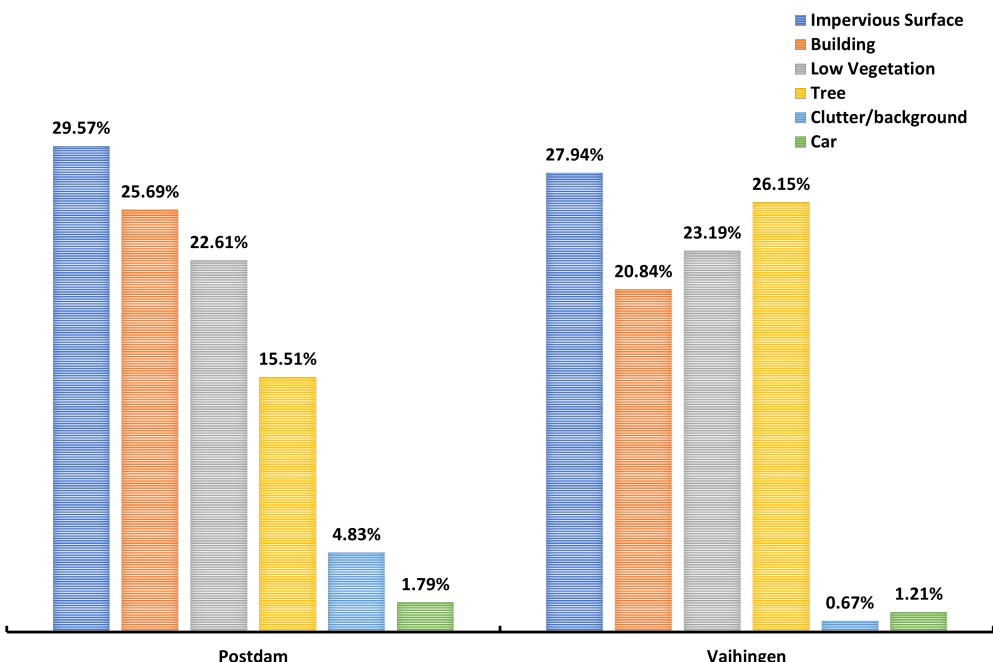

**Figure 5.** Proportion of each semantic label in the Vaihingen and Potsdam datasets.

### *4.2. Implementation Details*

We utilize a 12 GB 2080*Ti* GPU to train our model, employing the Adam optimization function and PyTorch as the programming language. Figure 6(3) illustrates our training procedure, which begins with pre-training the remote sensing segmentation model on the image dataset. This is subsequently followed by the primary training phase on the remote sensing dataset.

### 4.2.1. Pre-Training and Fine-Tuning Strategy

Our training strategy adopts a two-pronged approach. First, we employ the COCO dataset to pre-train our model, which takes approximately 30 h, utilizing a learning rate of

0.001 and the Adam optimizer. Subsequently, the model is fine-tuned on our remote sensing datasets. Model parameters stem from the pre-training, and we adjust the learning rate to 0.0001 to cater to the simpler nature of remote sensing images. This tailored approach ensures the model's proficiency in managing various occlusion cases and intricate object details pertinent to remote sensing. For data augmentation, we utilize random scaling, cropping, and both horizontal and vertical flipping to bolster the model's resilience to different scenarios.

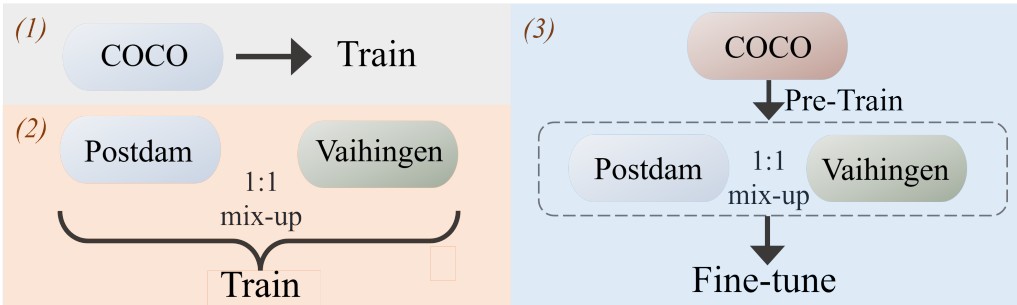

**Figure 6.** Our three different training strategies. Scheme 1 involves training only on a large-scale image dataset, specifically the COCO dataset. Scheme 2 involves training solely on remote sensing datasets, namely the Potsdam and Vaihingen datasets. The third Scheme is to perform pre-training on a large-scale image dataset and then conduct primary training on remote sensing datasets.

### 4.2.2. Model Configuration

The configuration of the Spatial-Aware Transformer and the Boundary-Aware Refinement module is specifically tailored to meet the unique requirements of remote sensing image segmentation. The Spatial-Aware Transformer comprises four stages, each with hidden dimensions of 96, 192, 384, and 768, and window sizes uniformly set at 7. The bottleneck layer is designed with 1024 channels. Training is executed in mini batches, each containing 16 images, and the process is carried out over 50 epochs.

### 4.2.3. Evaluation Metrics and Benchmarks

The efficacy of our model is primarily evaluated using *IoU* and boundary *F*1, which serve to validate the efficiency of both segmentation and boundary delineation. In addition, we employ Precision and Recall metrics to ensure a comprehensive assessment. Our performance benchmarks are established based on the current leading segmentation models for remote sensing, with a particular emphasis on models that have been trained on the Vaihingen and Potsdam datasets.

### *4.3. Ablation Study*

This paper requires ablation experiments on the Vaihingen dataset to verify the validity of the method and thus to discern the importance of each module.

### 4.3.1. Pre-Training and Main Training

Figure 6 presents three distinct training methodologies. The first approach involves training exclusively with the Potsdam datasets, the second employs only remote sensing image datasets, and the final method is a combination of pre-training and main training, which is the strategy we have adopted. Table 1 depicts the training accuracy corresponding to these three methods. An interesting observation is that the accuracy of the model trained solely with images surpasses that of the model trained only with video sets, underscoring the significance of dataset size in the training process. These experiments demonstrate that the abundant static image resources utilized in pre-training can contribute to enhancing the robustness of our network. Consequently, we employ a combination of pre-training and main training strategies to ensure that our model achieves optimal results.

**Table 1.** Ablation experiment of three different strategies on the Vaihingen dataset.

| Network Structure | MIoU (%) | Average F1 (%) |
|:---:|:---:|:---:|
| Scheme 1 | 65.26 | 74.57 |
| Scheme 2 | 64.31 | 71.92 |
| Scheme 3 | 74.88 | 81.44 |

### 4.3.2. Spatial-Aware Transformer Module

In order to evaluate the effectiveness of the Spatial-Aware Transformer (SAT) module, we conduct an ablation study where we systematically remove or modify certain components of the SAT and observe the impact on the model's performance. The goal of this study is to understand the contribution of each component to the overall performance of the model.

We start with a baseline model, which is the Swin Transformer without any modifications. We train this model on the same datasets and evaluate its performance.We modify the SAT to not include the spatial information in the Swin Transformer block. This will help us understand the impact of embedding spatial information into the Transformer block.

For each of these experiments, we measure the model's performance on the Vaihingen datasets using the same evaluation metrics. We compare the performance of each variant with the baseline model and the full SAT model to understand the contribution of each component. As shown in Table 2, with the help of the Spatial-Aware Transformer module, the $MIoU$ mean and average $\mathcal{F}1$ mean are improved by 6.5% and 1.4%.

**Table 2.** Ablation experiment of the Spatial-Aware Transformer on the Vaihingen dataset.

| Network Structure | MIoU (%) | Average F1 (%) |
|:---:|:---:|:---:|
| Ours | 74.88 | 81.44 |
| Swin Transformer [13] | 68.35 | 80.01 |
| Vision Transformer [33] | 66.35 | 79.13 |

### 4.3.3. Boundary-Aware Refinement Module

An ablation study was carried out to understand the contribution of the Boundary-Aware Refinement module in our Spatial-Aware Transformer (SAT). The study involves training three versions of the SAT: one with the Boundary-Aware Refinement module (referred to as SAT-B) one without a refinement module (referred to as SAT-NR), and one without the entire Boundary-Aware Refinement module (referred to as SAT-NB). The ablation experiments on the effect of the Boundary-Aware Refinement module were implemented on the Vaihingen dataset. The performance of both models is evaluated using the same metrics. Finally, we compare the performance of SAT-B, SAT-NR, and SAT-NB. As shown in Table 3, the improvement in the performance metrics indicates the effectiveness of the boundary-aware SAT-B module in the task of remote sensing image segmentation. The $MIoU$ mean and average $F1$ mean are improved by 2.9% and 1.8%. Figure 7 shows the comparison of the training accuracy of several different loss functions in our model training, and it can be seen that the training effect of our mixed loss function is the best.

**Table 3.** Ablation experiment of the Boundary-Aware Refinement module.

| Network Structure | MIoU (%) | Average F1 (%) |
|:---:|:---:|:---:|
| SAT-B | 74.88 | 81.44 |
| SAT-NR | 72.87 | 80.21 |
| SAT-NB | 71.92 | 79.76 |

### 4.4. Comparisons to State-of-the-Art Methods

4.4.1. Vaihingen Dataset

Table 4 shows a summary of the comparison between the different state-of-the-art methods, including the proposed methods, in terms of their performance on the aerial validation set. The results are sorted by IoU for impervious surfaces.The table includes eight different methods: Unet, Swin-UNet, FCN, TransUNet, Upernet, DANet, Deeplab V3+, and our method. Each method is evaluated based on seven performance metrics: IoU for impervious surfaces, buildings, low vegetation, trees, and cars, as well as the mean IoU and average F1 score.

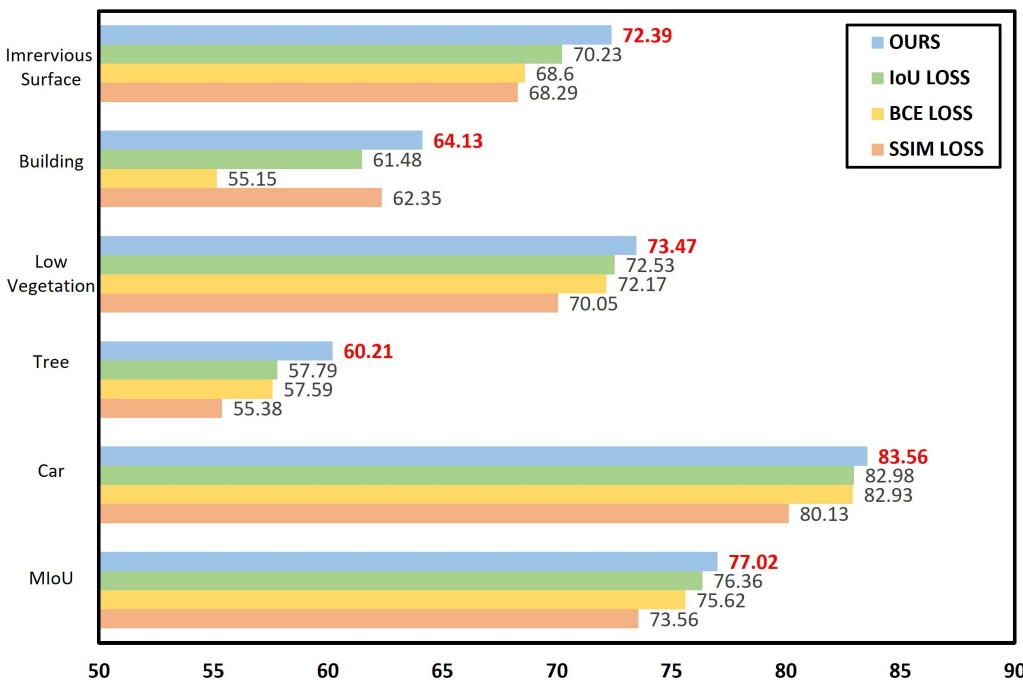

**Figure 7.** Ablation experiment of loss functions on the Vaihingen dataset.

**Table 4.** Comparison using Vaihingen validation sets. Results are sorted by IoU of impervious surface.

| | IoU (%) | | | | | Evaluation Index (%) | |
|---|---|---|---|---|---|---|---|
| **Method** | **Impervious Surface** | **Building** | **Low Vegetation** | **Tree** | **Car** | **Mean IoU** | **Average F1** |
| Unet [13] | 67.92 | 73.47 | 49.99 | 68.01 | 31.03 | 58.37 | 71.86 |
| Swin-UNet [4] | 72.03 | 82.46 | 56.97 | 72.02 | 58.30 | 66.46 | 80.01 |
| FCN [3] | 75.32 | 79.08 | 55.79 | 71.14 | 40.11 | 63.57 | 76.97 |
| TransUNet [34] | 73.58 | 80.86 | 54.97 | 70.98 | 54.63 | 68.33 | 80.03 |
| Upernet [36] | 72.91 | 80.99 | 55.89 | 72.24 | 47.31 | 64.74 | 79.06 |
| DANet [37] | 72.97 | 82.01 | 57.08 | 72.57 | 43.57 | 65.25 | 77.92 |
| Deeplab V3+ [38] | 73.95 | 84.33 | 55.86 | 72.65 | 51.43 | 67.27 | 80.69 |
| Ours | 77.73 | 82.39 | 75.38 | 77.59 | 61.31 | 71.60 | 81.44 |

A meticulous analysis of the table reveals that the proposed method, denoted as 'our' method, outperforms the other seven methods in terms of mean IoU (71.60%) and average F1 score (81.44%). This suggests a higher overall accuracy and harmonic mean of precision and recall, respectively, for the proposed method. Moreover, our method exhibits exceptional performance in the IoU of low vegetation (75.38%), which is significantly higher compared to the rest. For the IoU of impervious surfaces and trees, our method yields scores of 77.73% and 77.59%, respectively, indicating competitive results.

In contrast, methods like Unet and FCN have relatively lower scores across the board, suggesting their limitations in handling this particular dataset.

Figure 8 presents a comparative analysis of segmentation outcomes between our technique and other cutting-edge methods utilizing the Vaihingen dataset, depicted in Figures 1–3 and 5. It is evident that our approach yields more precise identification results.

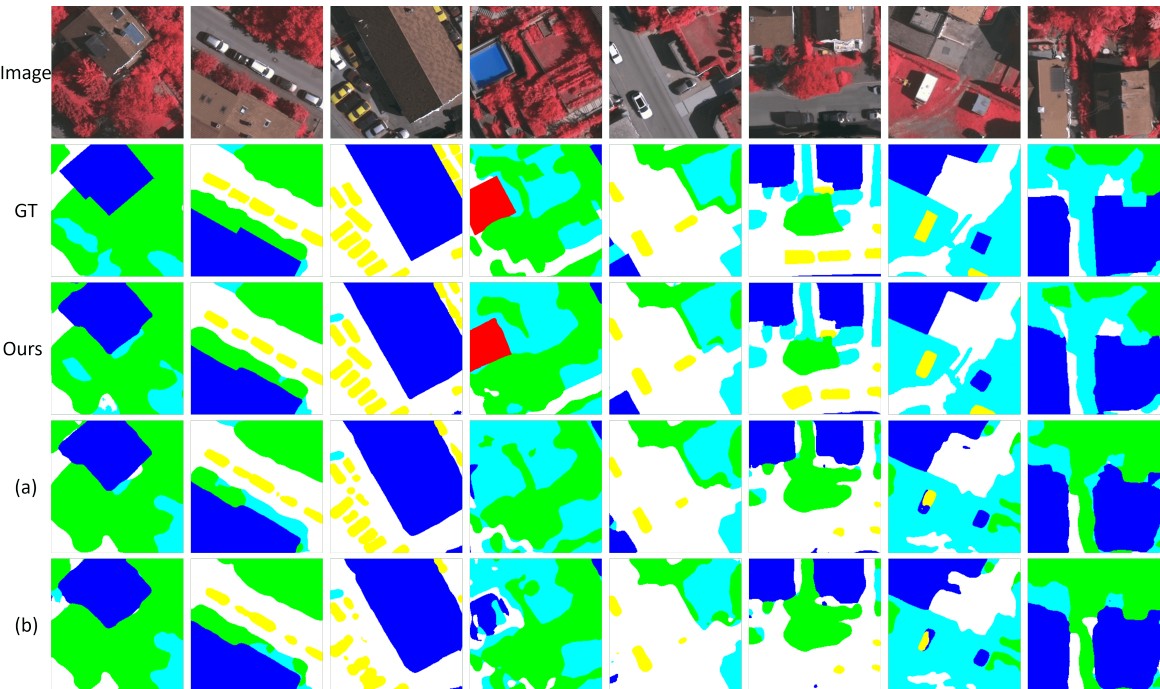

**Figure 8.** Comparison of segmentation results on the Vaihingen dataset. (**a**) represents the Swin Transformer and (**b**) represents the Vision Transformer.

This heightened accuracy can be attributed to a combination of pre-training on the COCO image dataset and the integration of our Spatial-Aware module, which together enhance our capability to discern different targets with a finer granularity. A case in point is the swimming pool in the fourth image, which is accurately segmented solely by our method. Furthermore, the incorporation of the Boundary-Aware Refinement module in our approach lends itself to achieving smoother object boundaries. This is particularly observable in the seventh image, where the segmentation is characterized by remarkably defined edges. This streamlined analysis not only maintains conciseness but also offers clear insights into the merits of our method in comparison to others, particularly in terms of target identification and edge optimization.

In conclusion, the proposed method demonstrates superiority in handling aerial validation sets, especially in extracting features of low vegetation, impervious surfaces, and trees with higher precision and reliability compared to other state-of-the-art methods.

### 4.4.2. Potsdam Dataset

Table 5 provides a comparative analysis of the segmentation outcomes for each method on the Potsdam dataset; it is discernible that our method surpasses all the other techniques with an average F1 score of 83.00%. This evidences a higher harmonic mean between precision and recall. Additionally, our method demonstrates a striking performance in the IoU of low vegetation, with a score of 76.23%, which is markedly superior compared to the others. In terms of IoU for impervious surfaces, our method excels with a score of 78.29%, while for trees it exhibits a highly competitive score of 77.10%. A notable observation is that Upernet performs exceptionally in the category of buildings with an IoU of 85.51%. Nonetheless, our method showcases a more balanced and consistent performance across

all categories. Contrarily, FCN appears to be less efficacious as per the table, especially in categories like Car IoU, where it scores only 30.83%.

Figure 9 illustrates the segmentation outcomes achieved by our method on the Potsdam dataset. Our method's efficacy in segmenting complex objects is commendable, which is a direct consequence of employing the Spatial-Aware Transformer algorithm. For instance, a close examination of the images in the second and fourth rows of the first column reveals intricate objects adeptly segmented. Moreover, our model excels in handling objects with convoluted edges, owing to the integration of the Boundary-Aware Refinement module. This is particularly evident in the images located in the first row of the third column and the fourth row of the third column, where the segmentation of objects with intricate edges is astutely precise. Figure 10 shows the segmentation performance of our algorithm in more complex environments, with images derived from the results of the Vaihingen dataset and the Potsdam dataset.

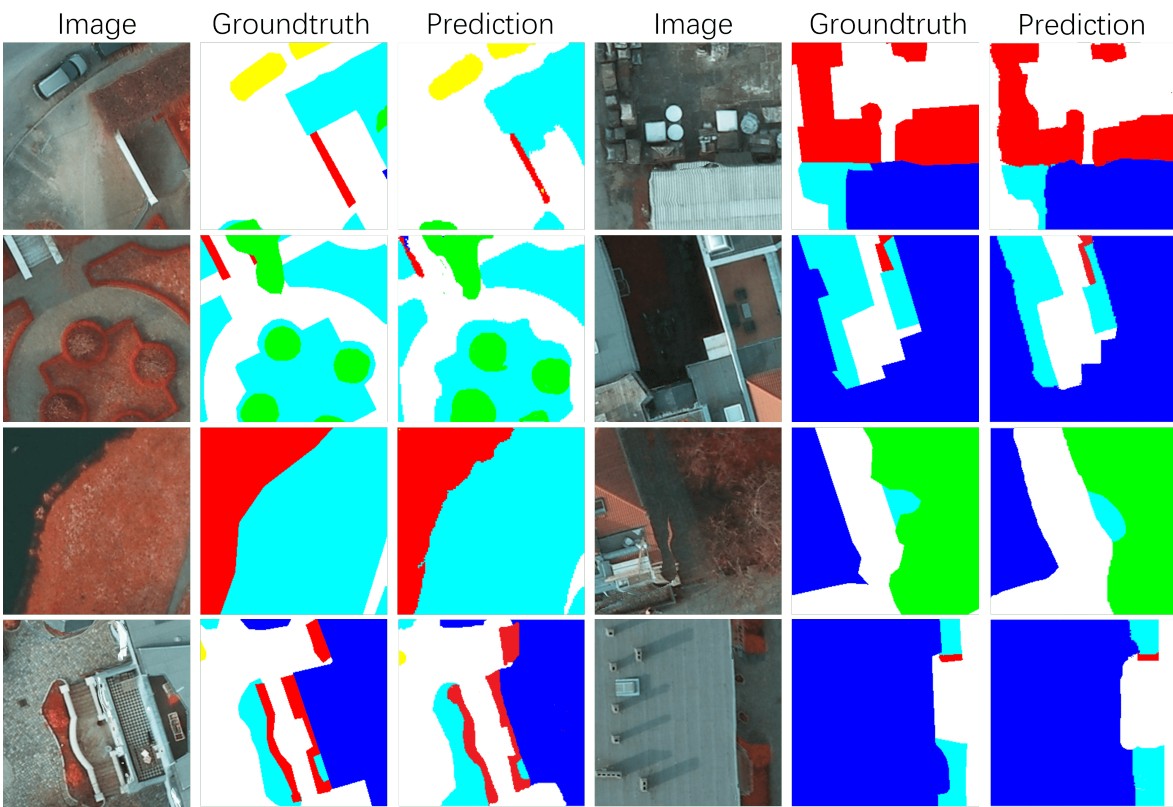

**Figure 9.** Comparison of segmentation results on the Potsdam dataset.

**Table 5.** Comparison using Potsdam validation sets. Results are sorted by IoU of impervious surface.

| | IoU (%) | | | | | Evaluation Index (%) | |
| Method | Impervious Surface | Building | Low Vegetation | Tree | Car | Mean IoU | Average F1 |
|---|---|---|---|---|---|---|---|
| FCN [3] | 67.32 | 74.74 | 50.65 | 67.68 | 30.83 | 58.28 | 71.53 |
| UNet [4] | 71.83 | 82.23 | 56.65 | 72.37 | 48.19 | 64.97 | 81.43 |
| Swin-Unet [13] | 75.80 | 79.00 | 54.36 | 70.04 | 42.53 | 63.15 | 77.16 |
| TransUNet [34] | 74.22 | 80.86 | 56.39 | 70.10 | 55.99 | 67.93 | 79.31 |
| Deeplab V3+ [38] | 73.00 | 82.25 | 56.17 | 73.33 | 46.49 | 64.09 | 80.32 |
| DANet [37] | 73.49 | 81.39 | 56.68 | 73.82 | 45.01 | 65.59 | 76.71 |
| Upernet [36] | 72.83 | 85.51 | 56.71 | 71.87 | 52.59 | 68.47 | 79.10 |
| Ours | 78.29 | 83.71 | 76.23 | 77.10 | 62.43 | 70.52 | 83.00 |

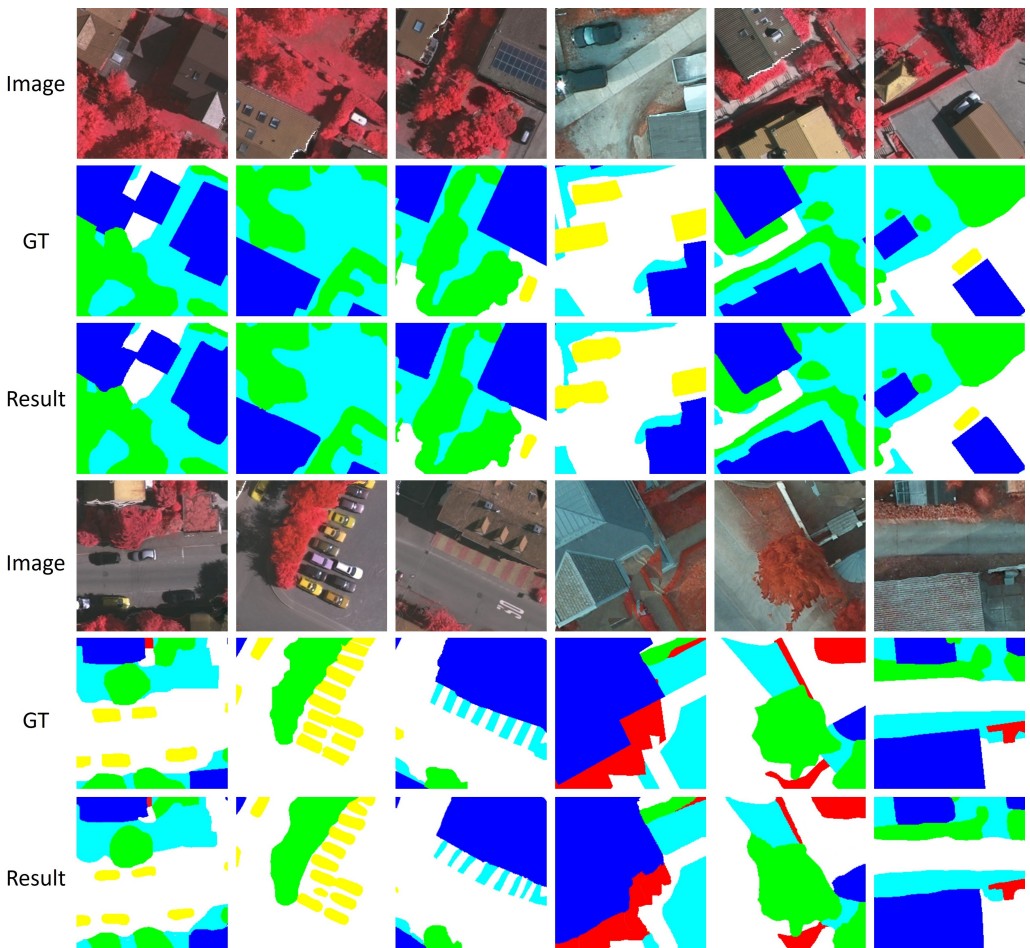

**Figure 10.** More results on the Vaihingen dataset and Potsdam dataset.

This analysis effectively communicates the key strengths of our method in dealing with complex objects and intricate edges, and presents the information in a coherent and concise manner.

In summary, the proposed method epitomizes robustness and reliability in dealing with Potsdam validation sets, rendering it an optimal choice for comprehensive and precise feature extraction. Moreover, its performance is particularly noteworthy in the evaluation of low vegetation, impervious surfaces, and trees.

### 4.4.3. Aerial Dataset

As shown in Table 6, it is evident that our proposed method outperforms the other approaches in terms of the mean Pixel Accuracy (mPA), mean Precision, mean Recall, and mean Intersection over Union (mIoU) for the roof category, with scores of 80.26%, 89.01%, 91.36%, and 98.49%, respectively. Notably, the mIoU for the roof category is exceptionally high, nearing perfection, which indicates an outstanding ability to correctly identify the relevant objects. In the context of the background category, our method also demonstrates superior performance. The mPA, mPrecision, mRecall, and mIoU are 97.91%, 97.26%, 98.00%, and 98.17%, respectively, which are amongst the highest scores in comparison to the other methods. Among the competing methods, Swin-UNet has the second-highest performance in the roof category in terms of mPA, with a score of 79.12%. For the background category, Swin-UNet also showcases remarkable results, especially in terms of mPA, mPrecision, and mRecall. It is crucial to mention that Swin-UNet achieves a notably high mRecall of 98.03% in the roof category, suggesting its proficiency in detecting relevant objects, though it does not lead in other metrics.

**Table 6.** Comparison using Aerial validation sets. Results are sorted by IoU of impervious surface.

| mIoU | Roof (%) | | | | Background (%) | | | |
|---|---|---|---|---|---|---|---|---|
| | mPA | mPrecision | mRecall | mIoU | mPA | mPrecision | mRecall | mIoU |
| Unet [13] | 72.39 | 83.41 | 81.36 | 86.34 | 81.85 | 87.89 | 91.39 | 96.21 |
| Swin-UNet [4] | 79.12 | 88.31 | 89.76 | 88.24 | 96.21 | 98.03 | 98.37 | 98.22 |
| FCN [3] | 76.52 | 81.01 | 83.93 | 81.59 | 80.32 | 84.09 | 96.39 | 97.13 |
| TransUNet [34] | 77.58 | 84.83 | 84.97 | 82.30 | 86.26 | 94.22 | 80.43 | 92.96 |
| Upernet [36] | 75.84 | 82.01 | 86.33 | 80.59 | 86.31 | 94.37 | 89.27 | 93.37 |
| DANet [37] | 76.77 | 82.50 | 86.37 | 81.00 | 81.32 | 96.30 | 87.16 | 94.21 |
| Deeplab V3+ [38] | 75.03 | 85.98 | 85.87 | 90.54 | 89.10 | 88.06 | 92.31 | 95.54 |
| Ours | 80.26 | 89.01 | 91.36 | 98.49 | 97.91 | 97.26 | 98.00 | 98.17 |

In Figure 11, the segmentation outcomes of our approach using the Aerial Imagery dataset are displayed. It is evident that our method yields both exceptional and consistent results. Furthermore, our algorithm demonstrates a remarkable ability to accurately segment dense building complexes, as is clearly illustrated in two instances—the second image in the first row and the image in the second column of the second row. This precise segmentation in complex scenarios highlights the robustness and efficacy of our method.

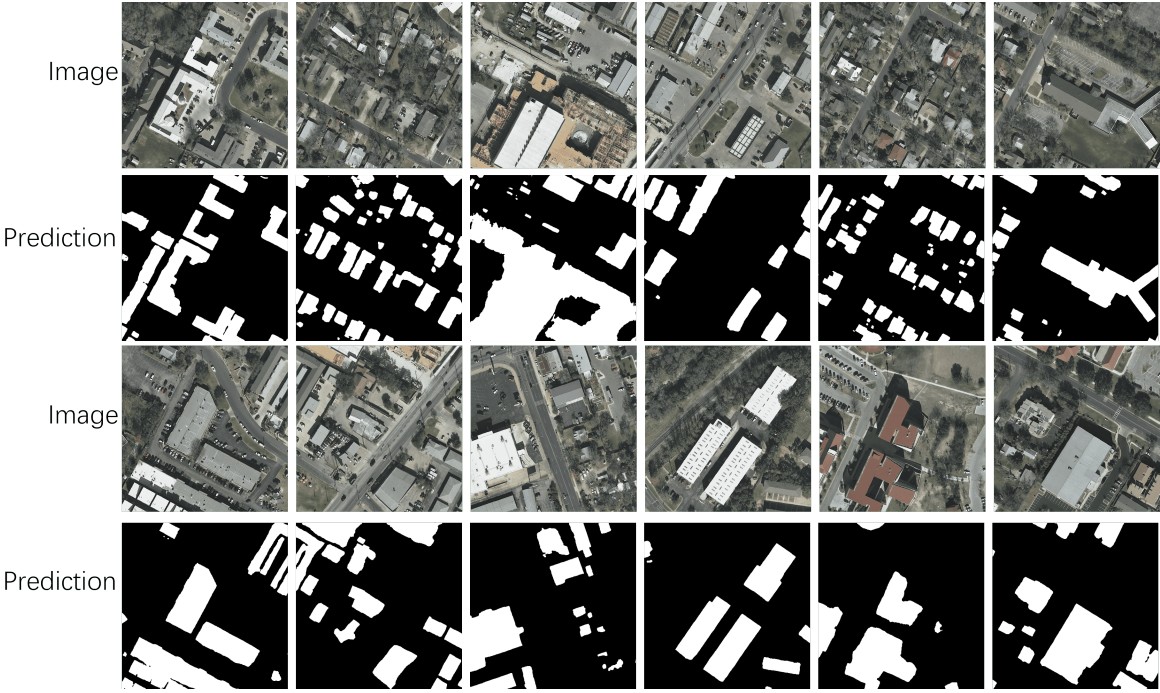

**Figure 11.** Comparison of segmentation results on the Aerial dataset.

In summary, our method evidently leads in performance across the board, demonstrating superiority in both the roof and background categories. This can be attributed to likely advancements in the architecture or training techniques employed. The table efficiently highlights these achievements and provides a clear comparative analysis among several state-of-the-art approaches.

### 4.5. Limitations

Our method, despite showing significant promise, acknowledges room for enhancement. As displayed in Figure 12, the segmentation results for elements like 'car' and 'road' are somewhat rough around the edges, indicating the need for more precise segmentation of smaller objects within the image. Throughout our experimental process, we identified in-

stances of semantic confusion appearing in certain segmentation outcomes. Such confusion typically manifests in images comprising categories that are semantically similar.

In conclusion, while our model may occasionally falter under specific extreme conditions, it consistently delivers high-quality segmentation results in the vast majority of cases. It proves especially adept at handling situations characterized by occlusion, distractions, and intricate object appearances. Furthermore, our network strikes an exceptional balance between accuracy and processing speed, underpinning its robust performance.

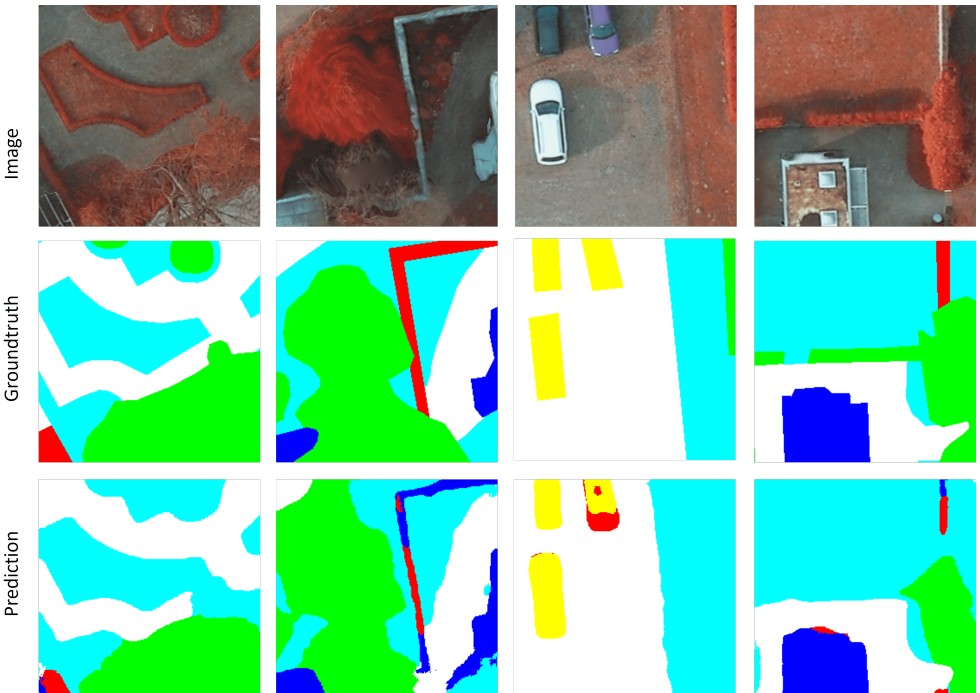

**Figure 12.** Unperfect results.

## 5. Conclusions

This paper presents a novel Spatial-Aware Transformer module and Boundary-Aware Refinement module, enhancing the Transformer segmentation algorithms for improved remote sensing image segmentation accuracy. The performance was evaluated using two remote sensing datasets: Vaihingen and Potsdam. The Spatial-Aware Transformer achieved a significant increase in the mean Intersection over Union (mIoU). Specifically, the mIoU reached 74.88% on the Vaihingen dataset and 75.55% on the Potsdam dataset. The Boundary-Aware Refinement module demonstrated superior accuracy in boundary segmentation. It achieved a Boundary $F_1$ score of 81.44% and 83.00% on the Vaihingen and Potsdam datasets, respectively. These results validate the utility of our innovative approach and demonstrate a significant potential for its application in remote sensing image segmentation tasks. However, additional research is required to handle more diverse and complex remote sensing tasks. Future work will focus on refining and expanding our method to further advance the state-of-the-art in remote sensing image segmentation.

**Author Contributions:** Conceptualization, D.W., Y.C. and B.L.; methodology, D.W.; software, D.W.; validation, D.W., Y.C., B.N. and B.L.; formal analysis, D.W.; investigation, D.W.; resources, D.W. and Y.C.; data curation, D.W.; writing—original draft preparation, D.W.; writing—review and editing, Y.C.; visualization, D.W.; supervision, Y.C., B.N., L.S. and B.L.; project administration, Y.C., B.N., L.S. and B.L.; funding acquisition, Y.C., L.S. and B.L. All authors have read and agreed to the published version of the manuscript.

**Funding:** This work is supported by the National Natural Science Foundation of China (Grant No. 61901191), the Shandong Provincial Natural Science Foundation (Grant No. ZR2020LZH005), and the China Postdoctoral Science Foundation (Grant No. 2022M713668).

**Data Availability Statement:** The COCO Dataset is available at https://cocodataset.org/#home (accessed on 21 February 2014). The Potsdam Dataset is available at https://www.isprs.org/ (accessed on 21 February 2015). The Vaihingen Dataset is available at https://www.isprs.org/ (accessed on 21 February 2015). The Aerial Imagery Dataset is available at https://captain-whu.github.io/AID/ (accessed on 21 February 2016).

**Acknowledgments:** The authors thank the anonymous reviewers and the editors for their insightful comments and helpful suggestions for improving our manuscript.

**Conflicts of Interest:** The authors declare no conflict of interest.

## Abbreviations

The following abbreviations are used in this manuscript:

| | |
|---|---|
| IoU | Intersection over Union |
| OA | Overall accuracy |
| SGD | Stochastic Gradient Descent |
| ReLU | Rectified linear unit |
| BCE | Binary Cross-Entropy |
| SSIM | Structural Similarity Index |

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
