# Peer review of "Spatial-Aware Transformer (SAT): Enhancing Global Modeling in Transformer Segmentation for Remote Sensing Images"

_remotesensing, doi:10.3390/rs15143607_

Round 1
Reviewer 1 Report
This paper presents a new deep-learning semantic segmentation method. The theory is correct and the results look good. I have some suggestions for it.
1. It is unclear what the f Fig.1 is trying to express. From the figure, it seems that Local Attention and Global Modeling are the same.
2. The author combines three loss functions and set three hyperparameters. However, in the ablation experiment, the authors did not conduct experiments on these three hyperparameters to analyze whether the hyperparameters’ value has an impact on the results.
3. LoveDA and GID are very challenging datasets in the remote sensing domain. They have wider coverage, more diverse ground cover scales, and inconsistent category distributions. It can better evaluate whether a model has better generalization ability and robustness. The authors should add experiments for these two datasets.
4. The last column in Table 6 is not given a name
The manuscript has minor language errors to be improved.
Author Response
See the attached file for details.

Reviewer 2 Report
The paper proposes a Spatial-Aware Transformer (SAT), an enhanced implementation of the Swin Transformer module for remote sensing image segmentation. It embeds spatial information into the Swin Transformer block and incorporates a boundary-aware module into the decoder to mitigate the shortcoming of inaccurate boundary segmentation. In general, the idea seems to be feasible, although the innovation is not so remarkable.
Below are my detailed comments:
1. This research focus on two primary components: the Spatial-Aware Transformer module, which enhances the global modeling capabilities of our segmentation algorithm, and the Boundary-Aware Refinement module, which sharpens the boundary segmentation in our output. Please further explain how the Spatial-Aware Transformer module make significant contributions to the field of remote sensing image segmentation tasks.
2. The model utilizes a conjoint loss function, incorporating BCE loss, SSIM loss, and IoU, Please give how the weights are assigned to each loss component loss?
3. In the Experiments, I suggest to supplement remote sensing image segmentation datasets in complex scenes for validation. The images used in current experiment is too simple.
4. Please provide how to accurately evaluate the effectiveness of the Boundary-Aware refinement module, and further demonstrate superior accuracy in boundary segmentation.
Author Response
See the attached file for details.

Reviewer 3 Report
This paper proposed an enhanced implementation of the Swin Transformer module, which includes two primary components: the Spatial-Aware Transformer module and the Boundary-Aware Refinement module. These modules cooperate to facilitate an effective approach to remote sensing image segmentation.
There are some points that I am concerned as follows:
(1) As the authors point out, the bottleneck layer is constructed from a pair of Spatial-Aware Transformer modules, but Section 3.3.1 describes bottleneck layer as a linear transformation. Please make a detailed explanation.
(2) The y and r in Eq. (5) should be separated to make it clear.
(3) In the Structural Similarity Index (SSIM) loss function, how do the authors choose the input images U and Uˆ?
(4) In line 367, the authors say that “Fig. 9 illustrates our training procedure, which begins with pre-training the remote sensing segmentation model on the image dataset.” It seems that the above description is not in accordance with Fig. 9.
(5) In Section 4.3.2, the Ablation experiment of spatial-aware transformer seems not very enough. On one hand, Vision Transformer aims to validate the importance of spatial information in the Swin Transformer block, where the spatial information is not an innovation of the authors. On the other hand, directly comparing Swin Transformer and “Ours” is not very fair. It is possible that increased network parameters promote the performance. To verify the effect of spatial-aware transformer, the authors should devise a basic network with similar parameter amount to replace the proposed attention structure in the upper part of Fig. 3.
(6) There are some errors in this paper, such as the dimension on line 266, the s^(t+1) in Eq. (4), etc. Besides, the i, j in Eq. (3) should be changed to b, a to be more clear.
(7) Please explain the (b)(c) in Fig. 8.
(8) The authors should unify the name of proposed module. It seems that “the Edge Optimization module”, “refinement module”, “optimization module” refer to the same module.
(9) In line 489, “This is particularly evident in the images located in the first row of the fourth column and the second row of the fourth column, where the segmentation of objects with intricate edges is astutely precise.” Please check the correctness of the index of columns.
(10) In Table 6, the metrics for Background are missing.
(11) In line 502-504, the metrics and numbers are not in accordance, please double check.
Author Response
See the attached file for details.

Round 2
Reviewer 1 Report
the authors clarified all my concerns. the quality of the manuscript may meet the requirements of possible publication
Author Response
Thanks for the positive comments, we will continue to doubel check the manuscript and make it better in the new version.
Reviewer 3 Report
(1) In 4.2. Implementation Details, “Figure 6 illustrates our training procedure, which begins with pre-training the remote sensing segmentation model on the image dataset. This is subsequently followed by the primary training phase on the remote sensing dataset.” “Figure 6” should be changed into “Fig. 6 (3)”.
(2) In Line 534-535, the metrics and numbers are not in accordance, please double check.
3) I’m not very clear about the explanation in Line 440-442. “We modify the SAT to not include the spatial information in the Swin Transformer block. This will help us understand the impact of embedding spatial information into the transformer block.” Does “modifying the SAT to not include the spatial information in the Swin Transformer block” correspond to the method “Vision Transformer”? If so, Vision Transformer has lower computational complexity than “Ours”. Please explain this difference of architecture between ViT and Swin Transformer.
Author Response
See the attached file for details